# Make It Less *difficile*: Understanding Genetic Evolution and Global Spread of *Clostridioides difficile*

**DOI:** 10.3390/genes13122200

**Published:** 2022-11-24

**Authors:** Mariachiara Mengoli, Monica Barone, Marco Fabbrini, Federica D’Amico, Patrizia Brigidi, Silvia Turroni

**Affiliations:** 1Microbiomics Unit, Department of Medical and Surgical Sciences, University of Bologna, 40138 Bologna, Italy; 2Unit of Microbiome Science and Biotechnology, Department of Pharmacy and Biotechnology, University of Bologna, 40126 Bologna, Italy

**Keywords:** *Clostridioides difficile*, PaLoc, antibiotic resistance, worldwide spread, ribotypes, genomic surveillance, genetic engineering

## Abstract

*Clostridioides difficile* is an obligate anaerobic pathogen among the most common causes of healthcare-associated infections. It poses a global threat due to the clinical outcomes of infection and resistance to antibiotics recommended by international guidelines for its eradication. In particular, *C. difficile* infection can lead to fulminant colitis associated with shock, hypotension, megacolon, and, in severe cases, death. It is therefore of the utmost urgency to fully characterize this pathogen and better understand its spread, in order to reduce infection rates and improve therapy success. This review aims to provide a state-of-the-art overview of the genetic variation of *C. difficile*, with particular regard to pathogenic genes and the correlation with clinical issues of its infection. We also summarize the current typing techniques and, based on them, the global distribution of the most common ribotypes. Finally, we discuss genomic surveillance actions and new genetic engineering strategies as future perspectives to make it less *difficile*.

## 1. Introduction

*Clostridioides difficile*, formerly known as *Clostridium difficile* [1], is an obligate anaerobic pathogen among the most common causes of healthcare-associated infections [2]. It was first described in 1935 by Hall and O’Toole, following isolation from a newborn’s stool, and initially called *Bacillus difficilis* due to the difficulty of cultivating it [3].

*C. difficile* is a Gram-positive, spore-forming, rod-shaped bacterium. Its spores are able to resist oxygen, heat, and common disinfectants (e.g., ethanol) thanks to their low water content and other properties, such as high levels of dipicolinic acid and saturation of DNA with soluble proteins [4,5,6,7].

This pathogen is transmitted via the fecal–oral route: if ingested, *C. difficile* spores withstand the acidic conditions of the stomach and then germinate in the intestine. This happens in the lower gastrointestinal tract where the oxygen concentration is lower [8]. Spore germination is triggered by certain compounds such as the primary bile acid, cholic acid, and cholesterol derivatives, including taurocholate. On the other hand, chenodeoxycholate is able to inhibit the germination of *C. difficile* spores [9,10,11]. Once germination has begun, the vegetative cells are able to colonize the colon and start producing toxins [12].

*C. difficile* generally produces two exotoxins, namely toxin A (TcdA) and toxin B (TcdB), which are both enterotoxic and cytotoxic. In addition, some *C. difficile* strains are capable of producing *C. difficile* transferase (CDT or binary toxin). These toxins damage the cytoskeleton of epithelial cells, causing breakdown of tight junctions, fluid secretion, and adhesion of neutrophils, thus leading to disruption of the intestinal barrier integrity, loss of function and local inflammation. In addition to toxin production, *C. difficile* virulence is also attributable to enzymes, such as collagenase, chondroitin-sulfatase, and hyaluronidase, which are capable of disrupting tight junctions as well, leading to fluid secretion and promoting inflammation [2,13,14].

With these prerequisites, the clinical picture of *C. difficile* infection (CDI) ranges from mild asymptomatic carrier status and diarrhea to fulminant colitis associated with shock, hypotension, or megacolon [15]. In the most severe cases of CDI, symptoms are critical (e.g., colon perforation, intestinal paralysis, septicemia) and can lead to death [15]. The mortality rate due to CDI is estimated at 5%, but the mortality associated with CDI complications ranges from 15% to 25% (up to 34% in intensive care units) [16,17,18].

Among all the potential risk factors for CDI, we can list increasing age (>65 years) and disorders such as inflammatory bowel disease, malignant tumors, diabetes mellitus and suppression of gastric acidity. However, the strongest risk factor for CDI is antibiotic therapy because it allows for the disruption of the gut barrier integrity [19]. In the study by Webb et al. [20], it is suggested that cumulative exposure to antibiotics, prior to admission for hospitalized patients, is the main contributor to CDI risk. Another risk concerns antibiotic classes: late-generation cephalosporins and carbapenems present a higher risk [21], but even relatively narrow-spectrum antibiotics, such as ampicillin, are associated with CDI. A recent work by Anjewierden et al. [22] has identified risks for asymptomatic CDI, such as hospitalization within six months prior to infection, nasogastric tube feeding, use of gastric acid suppression therapies, as well as use of corticosteroids in the previous eight weeks.

The gut microbiota is also known to play an important role in the pathogenesis of CDI. For example, the presence of bacterial consortia consisting of *Porphyromonadaceae*, *Lachnospiraceae*, *Lactobacillus*, and *Alistipes* might limit the growth of *C. difficile* [23]. On the other hand, in the case of dysbiosis exacerbated especially by the loss of primary fermenters, such as *Lachnospiridae* and *Lactobacillus,* a niche is established in which alternative primary fermenters such as *Bacteroides thetaiotaomicron* can proliferate. These microorganisms can produce various metabolic end-products that have been shown to be used by *C. difficile* as a carbon source [23,24,25]. Furthermore, some members of the gut microbiota are responsible for the generation of secondary bile acids, which, as anticipated above, can have an impact on the germination of *C. difficile* spore [23]. In particular, bacterial 7alpha-dehydroxylation of primary bile acids appears to protect against CDI [26], potentially representing a biomarker for a *C. difficile*-resistant environment. However, cefoperazone and other broad-spectrum antimicrobials have been shown to disrupt the bile acid-modifying activity of the gut microbiota, thus facilitating the colonization and outgrowth of *C. difficile* [27]. As described by Spigaglia et al. [28], the administration of certain antimicrobials (e.g., cephalosporins and carbapenems) is in fact more commonly associated with the induction of CDI than other pharmacological treatments.

One of the most common problems with CDI is recurrent CDI, which is generally associated with poor clinical outcomes. Some of the risk factors for recurrent CDI, recently described by Alrahmany et al. [29], are over 76 years of age, the total length of hospital stay (>7 days), previous exposure to clindamycin, and concomitant use of aztreonam.

Constituting one of the most important nosocomial infections, several guidelines have been developed to counter the spread of *C. difficile* [15,30]. However, ironically, the use of antibiotics is still considered first-line therapy. Of course, prolonged administration of antibiotics leads to the onset of antimicrobial resistance. The European Committee on Antimicrobial Susceptibility Testing (EUCAST) has established epidemiological cut-off values (ECOFFs), i.e., values that distinguish wildtype microorganisms from those with phenotypically detectable acquired resistance mechanisms (non-wildtype) to a given pharmacological agent [31]. According to EUCAST data, in some cases, *C. difficile* shows minimum inhibitory concentration (MIC) values above the ECOFF limit, especially against antimicrobials recommended by European and American guidelines, such as vancomycin, fidaxomicin and metronidazole [31]. However, fortunately, the resistance to these antimicrobials remains quite rare (see paragraph on “Antimicrobial resistance genes”) [32,33]. It should also be noted that a correlation between vancomycin treatment to counteract intestinal colonization of *C. difficile* and the acquisition of vancomycin-resistant *Enterococcus* (VRE) species has been demonstrated [34,35,36]. In particular, Fujitani et al. [35] found that the rate of VRE exceeded 50% in CDI patients, and patients with VRE had a higher rate of co-infection with multi-drug resistant pathogens. In an epidemiological study of hospital-acquired CDI, involving intensive care units, nearly one-third of patients with CDI were found to be colonized with VRE [37]. The use of intravenous vancomycin or metronidazole could be the main contributing factor to the risk of VRE colonization or infection [35].

Several studies focusing on both human and animal isolates have suggested a possible role for animals as sources of *C. difficile* [38,39,40,41,42,43]. A small Italian study, conducted on 39 samples, identified 14 different *C. difficile* ribotypes (RTs, see paragraph “Molecular typing techniques for *C. difficile* strains”), of which four (i.e., RT020, RT078, RT106, RT126) were detected in both animal and human samples. It should be noted that RT018 and RT078 were the two most frequently identified RTs, making up nearly 50% of all animal and human strains [44]. However, transmission of *C. difficile* can occur not only in a hospital setting or via animals. There is indeed evidence that *C. difficile* spores can also persist in the food chain, from farm to fork [45].

In light of the serious health implications, it is of the utmost urgency to better understand how to treat and contain the spread of this pathogen. The present review aims to investigate the genetic variation of *C. difficile* pathogenic genes and their relationship with the clinical issues of CDI. Then, we discuss the current typing techniques and the global distribution of the most common RTs, putting forward some hypotheses on the absence of hypervirulent RTs in some countries. Finally, this review summarizes actual genomic surveillance with a focus on novel genetic engineering strategies to make it less *difficile*.

## 2. Genetic Evolution of *C. difficile* Virulence

The *C. difficile* genome is characterized by a high level of plasticity. In particular, the evolution of its virulence concerns a specific genomic region, namely the Pathogenicity Locus (PaLoc). PaLoc consists of 19.6 kb of a chromosomally located element [46], which includes *tcdA* and *tcdB*, the genes encoding the two main toxins (TcdA and TcdB, respectively). PaLoc also comprises three accessory genes, namely *tcdR*, *tcdC* and *tcdE*. Specifically, *tcdR* encodes a positive regulator of toxin expression, while *tcdC* a negative regulator. *tcdE* encodes a putative holin-like protein, responsible for the secretion of microbial toxins [46,47]. Another toxin produced by *C. difficile* is the binary toxin (CDT), encoded by the Ctd Locus (CdtLoc), a distinct chromosomal region that carries *cdtA* and *cdtB*, the genes for catalytic and binding/translocation proteins, as well as *cdtR*, coding for a regulatory protein. The CdtLoc can be found in two versions, whole or truncated. In strains lacking CdtLoc, a unique 68-bp sequence is found inserted in the same genomic location [48,49]. In non-toxigenic strains, such as the recently discovered NTCD-035 by Maslanka et al. [42], PaLoc is replaced by a conserved non-coding sequence of 115 bp [46,50,51,52]. Ongoing works on the comparative genomics of *C. difficile* are highlighting some important features on the evolution of these two pathogenic regions. The entire PaLoc region appears to have a modular structure and its variability may be due to the substitution of single nucleotides and to recombination events that played a pivotal role in the evolution of PaLoc variants. This structure also affects the *tcdA* and *tcdB* genes. Interestingly, the CdtLoc region appears to be more conserved than the PaLoc region, but it is mainly observed that full-length CdtLoc is associated with *C. difficile* strains exhibiting significantly altered PaLoc [53]. Mansfield et al. [54] highlighted a difference between *tcdA* and *tcdB*: while the evolutionary history of *tcdB* may depend on extensive homologous recombination, *tcdA* shows a greater degree of sequence variation and a greater number of subtypes [54]. Therefore, the authors suggest that the extreme recombination events observed in *tcdB*, but not in *tcdA*, could lead to increased selective pressure for *tcdB* diversification, highlighting the potential role of *tcdB* in the pathogenesis of *C. difficile*. This observation seems to be confirmed also by studies focused on the use of monoclonal antibodies (e.g., bezlototumab and actoxumab) both in animal models (i.e., gnotobiotic piglets) [55] and in humans [54,56,57].

One of the few works so far focused on the evolutionary history of *C. difficile* is a study by Dingle et al. [52], which paves the way for understanding the genomic background of this bacillus. According to the authors, PaLoc acquisition occurred at separate times between the *C. difficile* clades. Based on multi-locus sequence typing (MLST), *C. difficile* strains can be stratified into at least eight phylogenetic clades: clade 1–5 and clades C-I, C-II and C-III [58]. Clade 1 includes over 200 toxigenic and non-toxigenic sequence type (STs). Clade 2 also contains several highly virulent strains (e.g., ST1). Little is known about clade 3, although it includes ST5, a toxigenic CDT-producing strain [58]. Clade 4 contains ST37, which is responsible for much of the endemic CDI burden in Asia [59] despite the absence of the *tcdA* gene. Clade 5 contains several CDT-producing strains (e.g., ST11), which are highly prevalent in production animals worldwide [60]. Clades C-I, C-II, and C-III, known as “cryptic clades”, were first described in 2012 [52,61] and contain more than 50 STs [52,61,62,63,64]; however, their evolution remains poorly understood [58,63]. In the study by Dingle et al. [52], the authors suggest that each lineage acquired its current PaLoc variant after the divergence and that the common ancestor of all modern *C. difficile* strains may have been non-toxigenic. In particular, clade 1, which includes the greatest diversity of toxigenic genotypes, may exemplify the most ancient acquisition; this fact would also explain the emergence of non-toxigenic strains within this clade, as sufficient time has elapsed for occasional PaLoc losses to occur. Moreover, the most recent PaLoc loss event occurred about 30 years ago within a genotype belonging to clade 1. In contrast, clades 4 and 5 exemplify the most recent PaLoc acquisition (about 500 years ago), because of their narrow genotypic diversity [52].

## 3. Antimicrobial Resistance Genes

One of the major issues related to CDI (and recurrent CDI) is antimicrobial resistance, which plays a crucial role in the pathogenesis and spread of *C. difficile* [33]. Resistance to specific antimicrobials may actually support the spread of *C. difficile*, as in the case of tetracycline resistance in RT078 [65] and clindamycin resistance in RT017 [66]. Although resistance to antimicrobials recommended by guidelines for the treatment of CDI remains quite rare [32,33], these guidelines have been modified and reshaped over the years, also to address this issue. In particular, the first guidelines (1995–1997) were focused on the administration of vancomycin and metronidazole, while from 2014 to date they also include treatment with fidaxomicin [67]. Fortunately, most countries in a pan-European study [32] reported metronidazole, vancomycin and fidaxomicin resistance below 10% (most of them reported 0%), and a species-wide genomic study [33] could identify pCD-METRO plasmid (conferring metronidazole resistance) in only 15 of > 10,000 strains tested. Below, we discuss the antimicrobial resistance mechanisms for medications commonly recommended for the treatment of CDI. Regarding the mechanisms of resistance, *C. difficile* strains can encode a vanG-type gene cluster (*vanGCd*), conferring resistance to vancomycin, an antimicrobial glycopeptide that inhibits bacterial cell wall biosynthesis [67]. Specifically, resistance is conferred by the production of an alternative lipid II carrying a D-alanine-D-serine terminus that is seven times less bound by vancomycin than the D-alanine-D-alanine terminus [68]. Constitutive expression of *vanGCd* has been shown in vancomycin-resistant strains isolated in the clinical setting as well as in laboratory-generated mutants, which carry mutations in the two-component VanSR system that regulates *vanGCd* [68].

Metronidazole-resistant *C. difficile* strains are occasionally isolated in clinical practice, but some strains appear to need the heme cofactor to be detectable [69,70]. The authors suggest that *C. difficile* may use heme in oxidative stress responses to metronidazole, as a source of iron and cofactor for redox-associated proteins, as shown in other bacteria [70,71]. Metronidazole belongs to the nitroimidazole class of antimicrobials that can block the helical structure of DNA, leading to strand breakage, inhibition of protein synthesis and cell death [67]. As anticipated above, resistance to metronidazole appears to be mediated by a high-copy-number plasmid, pCD-METRO [72]. However, so far, there are no data available as to which plasmid gene may actually be responsible for the resistance. Another genetic mechanism involved in metronidazole resistance in clinical isolates is the modification of the catalytic domains of pyruvate-ferredoxin/flavodoxin oxidoreductase (PFOR), a protein encoded by the *nifJ* gene [73]. This modification is mediated by a synonymous codon change in putative xanthine dehydrogenase, which is likely to affect mRNA stability, and frameshift and point mutations that inactivate the iron-sulfur cluster regulator [73].

Another drug commonly used to treat CDI is fidaxomicin, a narrow-spectrum macrocyclic antimicrobial that inhibits RNA synthesis. This drug acts through lipiarmycin A3 (Lpm), its active component, which inhibits bacterial RNA polymerase [74]. Hence, resistance to fidaxomicin is conferred by mutations affecting RNA polymerase, especially the β-subunit of the RNA polymerase [75,76]. Alternative therapies for the treatment of CDI recommended by the European Society of Clinical Microbiology and Infectious Diseases (ESCMID), the Infectious Diseases Society of America (IDSA), and the Society for Healthcare Epidemiology of America (SHEA) [30,77] include rifaximin and tigecycline. Like fidaxomicin, rifaximin, which belongs to the rifamycin family, inhibits bacterial RNA polymerase, especially the β subunit [28,78]. The β subunit appears to be the primary site of mutations responsible for modifying the rifamycin binding pocket, thus limiting the interaction between the target and the antimicrobial [79]. Tigecycline, a glycylcycline chemically and structurally similar to tetracyclines, has been proposed for the treatment of CDI due to its ability to inhibit spore formation and reduce the toxin production by the pathogen [80]. It should be noted that some *tet* genes (i.e., *tet*(X3) and *tet*(X4), encoding a flavin-dependent monooxygenase), which [81,82,83,84,85,86,87] typically confer tetracycline resistance, also appear to mediate tigecycline resistance in various microorganisms of clinical relevance [81,82], thus challenging the clinical efficacy of the entire family of tetracycline antibiotics, including any derivatives.

## 4. Molecular Typing Techniques for *C. difficile* Strains

Several molecular methods are available for typing *C. difficile*, and routine typing is not performed with the same techniques across countries. The most widely used *C. difficile* typing technique is PCR ribotyping. Specifically, this technique consists in the amplification of the intergenic spacer region (ISR) of the 16S-23S rRNA gene using primers targeting the 3′ end of the 16S rRNA gene and the 5′ end of the 23S rRNA gene [83]. Due to its high discrimination capacity, PCR ribotyping is currently recommended by the European Centre for Disease Prevention and Control (ECDC) for surveillance of the *C. difficile* spread [84]. However, this molecular technique has a delivery time of up to one week and often requires in-house protocol optimization, thus not being fully transferable between laboratories [85]. Another technique in use is pulsed-field gel electrophoresis (PFGE). This technique is based on DNA digestion, through enzymatic restriction, and separation of DNA fragments on gel. This method therefore provides for the clonal assignment of the bacterial strain based on PFGE banding patterns [86]. The criticality of this method can be identified in the tendency of *C. difficile* DNA to rapid degradation, resulting in non-typable isolates. To overcome this problem, a modified PGFE method has been designed, with different plate cultures compared to the standard protocol, but its application is rare in light of the other methods adopted for molecular typing [87]. Among these, restriction endonuclease analysis (REA) is a technique based on the use of restriction enzymes (e.g., HindIII) but, unlike PFGE, the digestion fragments obtained are separated by classical electrophoresis on agarose or polyacrylamide gels [83]. Other noteworthy typing methods are Multilocus VNTR Analysis (MLVA) and MLST. Specifically, the target of MLVA are variable number tandem repeats (VNTR), disseminated throughout the genome, while MLST uses PCR amplification of housekeeping genes to generate a complete allelic profile [88].

Since there are a multitude of different typing techniques, a web-accessible database [89] (always updated) was set up in 2011 by Griffiths et al. [90]: they typed a total of 49 isolates by MLST and classified them into 40 STs. Since MLST and PCR ribotyping are very similar in discriminatory abilities, they found a correspondence between RTs and STs: multiple RTs for the same ST and multiple STs for the same RT usually had very similar profiles. Some STs correspond to a single RT (e.g., ST54/RT012), while others to multiple (e.g., ST02/RT014, RT020, RT076 and RT220). However, it should be noted that RTs were not always predictive of STs [90].

In recent years, with the advancement of whole genome sequencing (WGS) techniques, the scientific community is moving towards these methods instead of standard PCR ribotyping and is trying to develop new methods for the characterization of *C. difficile* strains [85,91]. The two main approaches to discover genomic variations are single nucleotide variant (SNV) analysis and core genome or whole genome MLST (cgMLST, or wgMLST). The first technique is based on comparing the differences in single nucleotide polymorphisms (SNPs) while the second is based on the analysis of multiple genes across the whole genome. Cg- or wgMLST typing works according to the same principles as the classic MLST [91]. Three publicly available schemes for *C. difficile* are available for cg- and/or wgMLST typing, and analysis can be performed using commercial software (e.g., BioNumerics, Ridom) or freely accessible online resources (e.g., EnteroBase). Eyre et al. [92] were the first to use WGS of *C. difficile* genomes on benchtop sequencing platforms to investigate its transmission, demonstrating that the use of these technologies could improve infection control and patient outcomes in routine clinical practice. Since then, WGS typing has been widely adopted for CDI surveillance and has revealed some novel insights concerning the spread of *C. difficile* [91].

## 5. Worldwide Distribution of *C. difficile* Ribotypes

In the following chapters, we will offer a glimpse into the global distribution of the *C. difficile* RTs, divided by country. These data are summarized in Figure 1, while the toxin gene profile of the mentioned RTs is shown in Appendix A.

### 5.1. Europe

In Europe, several studies have highlighted a common genetic profile of *C. difficile* [93,101,102,103]. In particular, in 2021, a global study by Zhao et al. [101], based on data obtained from the MODIFY I (NCT01241552) and MODIFY II (NCT01513239) clinical studies using a WGS approach [104], found that in Europe there is a predominance of clade 1, with the exception of Poland where clade 2 predominates. Clade 2 was found to be one of the most virulent, along with clade 5. Interestingly, these two hypervirulent clades show the lowest recombination rates, while clade 3 and clade 4 show similar recombination rates [101]. Regarding the classification into RTs, clade 1 includes the non-toxigenic RT009, RT010, and RT039 [105], while the hypervirulent RT027 belongs to clade 2 [101]. The study conducted by Zhao et al. offers a broad view of the *C. difficile* genotype distribution at the global level but lacks comprehensive data. In fact, the study took into consideration only 1501 clinical isolates distributed globally. Another less recent but more accurate study analyzed 3499 isolates from 40 sites across Europe [93]. This study describes the 2011–2016 epidemiological framework. In particular, there was a well-defined predominance of RT027 during the five years, followed by RT001 in 2011 and RT014 in 2012, 2013 and 2014 (both RTs are toxigenic). In 2015, a predominance of RT014 was observed, followed by RT106 and RT002 [93]. These results appear to be in contrast with the observation by Zhao et al. [101], but they are probably more accurate due to the greater number of isolates analyzed. It is also worth mentioning the two works by Abdrabou et al., who considered Germany in the periods 2014–2019 and 2019–2021 [102,103]. The authors noted that in the first period there was a prevalence of RT027, with a decrease in the following years [102,103]. As discussed by the European Medicines Agency (EMA), the Centers for Disease Control and Prevention (CDC) and the U.S. Food and Drug Administration (FDA), such a decrease could be due to a potential reduction in fluoroquinolone administration [106,107,108]. Fluoroquinolones have in fact been associated with CDI outbreaks with RT027, which is highly resistant to them [109,110].

Similarly, comparing these results with other studies, a decrease in the prevalence of RT001 in Germany over time was observed [102,111,112]. It is also interesting to note that certain foods, such as potatoes, could be a vector for the introduction of *C. difficile* spores into the food chain or household environment. Indeed, potatoes have the highest rates of *C. difficile* contamination tested to date [113,114]. A recent study analyzed the positivity rate and distribution of RTs on potatoes in 12 European countries in the first half of 2018. Thirty-three of the 147 samples tested were positive for *C. difficile*, and the most common RTs were RT126, RT023, RT010, and RT014, in part overlapping what was discovered for human samples. The multiplicity of RTs was found to be substantial and the overlap between countries moderate.

### 5.2. America

In the USA, clinical specimens acquired from 2011 to 2015 showed a prevalence of RT027, belonging to the hypervirulent clade 2, as well as a prevalence of clade 1 [101]. Interestingly, the prevalence of clade 2 was higher on the East Coast and West Coast than inland. This observation was reversed for clade 1, which was more prevalent inland [101]. Another recent study [94] focusing on stool specimens recovered from 2011 to 2016 in the states of Illinois, Minnesota, New York, Massachusetts, California, and Virginia showed a marked decrease in the prevalence of RT027 over these six years in agreement with what was seen in Europe and Canada [94,112,115,116,117,118]. This decrease was also found in a 2020 study [119], covering the 2011–2018 period in Texas, where the most common RT was RT027, followed by RT014-020, RT106, and RT002. Curiously, the authors found a novel emerging RT, RT255 [119]. The complete RT255 genome has recently been made publicly available [120] and has occasionally been isolated [94]. However, its attributes and associated clinical outcomes are not yet well described [119].

Considering the transmission of *C. difficile* between humans and animals, an interesting study of samples collected from 13 Ohio swine farms (from farrowing rooms, nursery rooms and workers’ breakrooms) showed high contamination with toxigenic *C. difficile* [121]. The same *C. difficile* RTs recovered from most of the farm breakrooms were also recovered from at least one swine environment in those same farms. Furthermore, three RTs (i.e., RT078, RT005 and RT412) identified in the environment had previously been found in association with CDI in humans [122,123] and with animal-to-human transmission in Europe (i.e., Italy) [44]. Water and soil are also an important reservoir for this pathogen. At the Flagstaff site (Arizona), researchers found potential novel strains belongings to clades C-I, C-II and C-III and a hypothetical additional clade (C-V) [124].

In contrast to the literature describing the situation in North America, few recent articles are available on what concerns South America, mainly from Brazil. Regarding the latter, a study using MLST found that patient stool samples from different hospitals were positive for *C. difficile* distributed in 14 STs [125]. In particular, it was the first description of the STs 15 and 54 in the Brazilian country. ST15 has already been described in the UK [90], but as it is a non-toxigenic ST, there are not many data in the literature, while ST54 is already widespread in South America [125]. Unlike Europe and North America, the RT027 epidemic has never been reported in Brazil and the epidemiology of CDI is still underexplored. This is partly due to the lack of specialized technologies and facilities for the detection of obligate anaerobic bacteria, which is not a routine procedure in clinical laboratories [126]. However, numerous *C. difficile* RTs involved in CDI cases have been detected in Brazilian hospitals (e.g., RT014, RT043, RT046, RT106, RT132-135, RT142, and RT143) [96,126,127,128].

### 5.3. Asia and Middle East

While for Europe and North America there are numerous studies on the genetics and epidemiology of *C. difficile* strains, Asia appears to be split in half. For example, there is a lack of scientific reports on the prevalence of *C. difficile* in South Asian countries such as India, while these works appear to be abundant in East and Southeast Asian countries such as China and Japan. In India, a recent study using MLST on samples from a tertiary care center found that the most common STs were ST17, ST54, and ST63, with ST17 being the most prominent [129]. In Bangladesh, the first report on the prevalence of CDI in the hospital setting found that the most common RTs in stools (i.e., present in ≥10% of isolates) were RT017, RT053–163 (the same RTs found in environmental isolates), and a new RT (i.e., FP435) [130].

Furthermore, another study conducted in 2020 on the antimicrobial susceptibility of *C. difficile* in the 2014–2015 period in Asia-Pacific countries highlighted the prevalence of RT017 [95], confirming what was observed in a previous study [130]. Interestingly, the hypervirulent RT027 and RT078 have only rarely been isolated in the Asia-Pacific region [131,132,133,134]. In Japan, evidence suggests that the most common RT is RT018 [133,134,135,136]. In China, one of the dominant circulating RTs is RT017 [59,137,138]. However, another study published in 2021 on samples from different sources (e.g., soil, animals) found that the predominant RTs in China are RT001, RT046, and RT596 [139]. In 2021, the study by Zhao et al. found that the most common clade in the Asian country is clade 4 [101]. Furthermore, another report focusing on antibiotic resistance and molecular features of economic animals in China showed that RT126 is the most prevalent in Shandong province [140]. As for the Middle East region, RT001 is the most prevalent in Iran, followed by RT126, while RT258 is widespread in Qatar, RT139 in Kuwait and RT014 in Lebanon [141,142,143,144].

### 5.4. Oceania

Most of the data on the incidence of CDI come from Australia, the largest country in Oceania, particularly Western Australia. However, probably due to the variety of patient characteristics such as age, there is a discrepancy between the studies available to date.

For example, a study of *C. difficile* isolated from pediatric patients hospitalized in Perth, capital of Western Australia, in the period 2019–2020, reported a prevalence of RT002 (toxigenic) and RT009 (non-toxigenic) (on a total of 427 stool samples) [98]. On the other hand, a study conducted in the earlier period 2013–2018 in a geographically larger area covering 10 diagnostic laboratories from five states in Australia (Western Australia, New South Wales, Victoria, South Australia, and Queensland), reported 203 different RTs in predominantly elderly subjects, with RT014/020 being the most common, while RT027 and RT078 only rarely found [145]. Additionally, toxigenic RT014/020 was found to be more common among clinical cases in a tertiary hospital in Perth, while non-toxigenic RT010 was prevalent among floor samples and shoe soles of hospital staff, visitors and patients [146]. Outside of Australia, few research works are available. In particular, for New Zealand, a 2021 study reported that the two most common RTs were the same found in Australia (i.e., RT014, RT020) [147]. Interestingly, the dominance of RT014 has also been reported in Europe, particularly in 2015 [93]. The dominance of RT014 was also found in Auckland in 2014 [148]. In the study by Zhao et al., clade 1 was found to be the predominant one in Oceania [101].

### 5.5. Africa

In Africa, *C. difficile* is generally considered a minor pathogen, while the most common causative agents of diarrhea are *Escherichia coli*, *Cryptosporidium*, *Shigella,* and rotavirus [149]. No less important, there is a lack of recent literature on ribotyping and molecular characterization of *C. difficile*. For example, for Northern Africa, the latest PCR ribotyping report was released in 2018 in two Algerian hospitals, revealing the predominance of RT014, RT020 and non-toxigenic RT084, but only in 11 out of 159 stool samples collected between 2013 and 2015 [99]. In sub-Saharan Africa (i.e., Tanzania), a 2015 study identified RT038 among non-toxigenic strains, RT045 among toxigenic strains, and an unknown RT for two strains resulted positive for both *tcdA* and *tcdB*. Further analyses conducted on one of the two unknown strains highlighted a similarity with RT228 and RT043 [100]. Another study conducted in rural Ghana showed a high rate of non-toxigenic strains (e.g., RT084) isolated from patients with diarrhea and no hypervirulent strains [150]. A 2018 multi-centric cross-sectional study conducted between Germany, Ghana, Tanzania, and Indonesia showed that non-toxigenic strains were more abundant in Africa, with a prevalence of RT084 in the Ghanaian site and RT038 and RT045 in the Tanzanian site [151]. In light of this evidence, there appears to be a shortage of hypervirulent RTs in Africa (as opposed to Europe and the USA), while most of the CDIs found are attributable to non-toxigenic strains. Finally, it should be noted that in the Zimbabwe region, RT084 was the most common in human samples, while RT103, RT025, and RT070 were prevalent in chicken isolates, and RT025 and RT070 in soil samples [152]. Based on the study by Zhao et al., in South Africa, there is a prevalence of clade 1 [101].

## 6. Future Perspectives to Make It Less *difficile*

### 6.1. Genomic Surveillance

Recently, the COVID-19 pandemic has exposed the challenges for genomics in public health surveillance systems, given the rapid spread of new viral variants [153]. Genomic surveillance is transforming public health action by providing a deeper understanding of pathogens, their evolution and circulation. As mentioned earlier, new technologies in sequencing and bioinformatics have recently emerged and we have now reached a point where genomic surveillance has a clear role to play for public health. Specialized laboratory techniques such as WGS are increasingly used according to the WHO Global Surveillance 2022 report, in the investigation and acute management of diseases that could constitute public health emergencies, including cholera, influenza, Ebola virus disease, bacterial meningitis and poliomyelitis [154]. Public health responsiveness to pathogens could also benefit from larger-scale genomic surveillance, for example to monitor the occurrence and spread of *C. difficile* variants. However, for this to happen, several actions would be required. First, there is a strong need to improve access to tools for a better geographical representation of the pathogen spreading, sampling not only hospitalized patients with symptoms, but also environmental hotspots that can be enriched in human-derived pathogenic microorganisms, such as wastewater. Recently, sewage systems have gained increasing attention for health surveillance purposes due to their relevance in improving responsiveness to SARS-CoV-2 infection peaks, enabling early detection of viral loads and the emergence of new viral variants [155,156]. *C. difficile* is commonly found in raw sewage and survives the wastewater treatment process, thus being disseminated into the environment via effluent and land application of biosolids [157]. A genomic investigation conducted by Moradigaravand et al. [158] in 65 patients showed that CDI was attributable not to hospital-derived spores, but rather to strains isolated from effluent water from nearby treatment facilities, bolstering the need for bacterial genomic control in the environment [158]. This first action implies the need to strengthen the workforce to support this monitoring burden between countries and improve data sharing in a local-to-global fashion, enabling shared decision-making. Furthermore, the production of big data on a global scale encompassing genomic data, geographic origin, and spread of the isolates in the population will serve as a starting set to train a genomic-informed, real-time, global pathogen predictive model that might allow for greater control of infection outbreaks and conscious management of wastewater and other pathogen reservoir hotspots. Surveillance, outbreak detection, and response fully adhere to the One Health concept, which encompasses the domains of human, animal, and environmental health [159]. A deep global genomic characterization of *C. difficile* isolates could definitely help improve not only the profiling, but also the treatment of infected patients in a more rational, knowledge-based way.

### 6.2. Genetic Engineering

In recent years, a second-line approach leveraging synthetic biology is receiving particular attention, as an innovative strategy to limit *C. difficile* outbreaks by reducing its pathogenicity. Indeed, a wide range of tools are now available to precisely delete genes, change single nucleotides, complement deletions, integrate novel DNA, or overexpress genes [160]. Among these, the CRISPR/Cas system is of great interest for its ability to produce targeted modifications with high precision and efficiency, and to act on multiple strains [161]. CRISPR-mediated genome editing has been successfully implemented to tackle *C. difficile* infectious process, by hijacking the native system to introduce selective knock-outs [154] of several virulence factors involved in its pathogenicity in humans, deepening our knowledge over certain pathogenic processes [162]. For example, the study of the flagellin FliW-CsrA-*fliC*/FliC regulation mechanism implementing a CRISPR-based deletion of *fliW, csrA* and *fliW-csrA* genes demonstrated that the RNA-binding protein CsrA negatively modulates *fliC* expression, whilst FliW indirectly affects *fliC* expression through inhibition of CsrA post-transcriptional regulation [163].

Understanding *C. difficile* virulence regulation and pathogenicity factors through gene editing techniques also paves the way for efficient attenuation and possible therapeutic strategies. For example, toxicity and early-stage adhesion can be counteracted by knocking-out the *tcdA* and *cpw84* genes, respectively [164]. While *tcdA*, as previously mentioned, encodes toxin A, *cpw84* encodes a cell wall protein, the main protease of *C. difficile*, important in the early stages of colonization [164]. Another example of effective attenuation of *C. difficile* virulence was proposed by McAllister et al. [165], who produced a CRISPR-Cas9 mutant of *C. difficile* lacking the selenophosphate synthetase gene *selD*, resulting in a growth-deficient mutant [165]. Similarly, Wang et al. [166] achieved the complete deletion of the sporulation protein spo0A in a wildtype strain of *C. difficile*, undermining its ability to survive environmental stresses. Finally, Selle et al. [167] proposed an interesting delivery system for in vivo targeting of *C. difficile*, by developing a phage-delivered CRISPR/Cas3-mediated antimicrobial [167]. Phage infection was selective and apparently safe, activating a repurposing response of the endogenous CRISPR/Cas system that induced self-targeting endonuclease activity.

Such synthetic biology studies pave the way for new treatments in the field of personalized medicine, with in vivo strain-specific targeted genomic modifications to make *C. difficile* more vulnerable to therapies, thus providing the patient with a rapid and relapse-free recovery. Furthermore, such strategies can be virtually translated to other bacterial infections and could provide powerful and reliable strategies for overcoming the challenges of infections with multidrug-resistant microorganisms directly in the context of complex microbial communities in vivo.

## 7. Conclusions

This review gives a glimpse on the worldwide spread of *C. difficile* from a genetic standpoint. *C. difficile* is nowadays a global threat due to the clinical outcomes of its infection and growing concern about resistance to antimicrobials. From a methodological point of view, it should be remembered that *C. difficile* is not easy to grow and, although the ECDC guidelines suggest using PCR ribotyping as the main typing technique, many studies from different countries have been conducted with other methods, obviously challenging the overlap of results. Additionally, typing data are absent in Northern and Central Africa or some Asian countries, making the view of what *C. difficile* burden really represents even more fragmented. However, to date, as far as we know, there is a majority of toxigenic RTs in Western countries (e.g., RT027 in Europe), as well as in Asian countries (e.g., RT017), while in Africa, the most abundant strains are non-toxigenic (e.g., RT084). Failure to detect hypervirulent RTs could be explained by multiple factors, related to populations and their lifestyle, as well as poor typing effort. For example, as regards the African continent, CDI patients are often younger than in European and North American contexts, but typing data are rarely reported [168]. Furthermore, co-infection with the causative agents of tuberculosis or malaria has been reported to affect the virulence of *C. difficile* RTs [169], although the underlying mechanisms are unknown. Finally, the low availability of antimicrobials in some Asian and African countries and the suggested reduction in their use by international agencies [170,171] could certainly be linked to lower selective pressure on circulating *C. difficile* strains, potentially helping to foster the emergence of non-hypervirulent RTs.

Another relevant issue in CDI management is the lack of data on non-clinical transmission, i.e., through the food chain or the environment. In addition to a homogenization of methodological approaches, future studies should therefore aim at a better sampling and geographic representation of the *C. difficile* spread, in order to better understand (and interfere with) its transmission dynamics. With specific regard to therapies, in recent years, genome editing technologies such as the CRISPR/Cas system are demonstrating their potential in improving the susceptibility of *C. difficile* to treatments, thus opening up unprecedented opportunities for personalized medicine. Coupled with the mapping effort, advances in genomics and bioinformatics could lead to a better understanding of what this pathogen is and, more importantly, how to make it less *difficile*.

## Figures and Tables

**Figure 1 genes-13-02200-f001:**
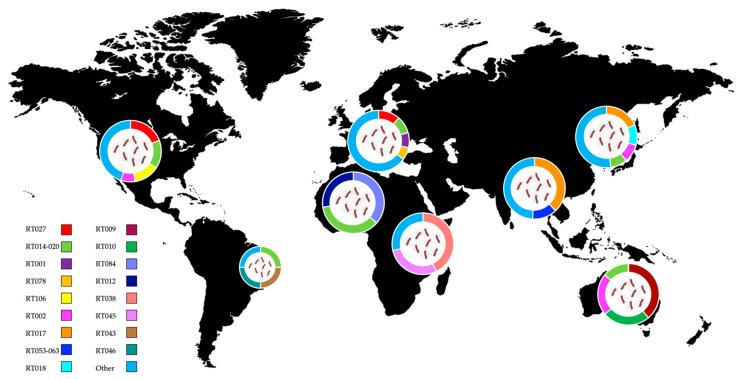
Distribution of *C. difficile* ribotypes in the United States, South America, Europe, North and Central Africa, Asia and Australia. Data are shown in pie charts as percentage, from studies covering different countries [93,94,95] or single country [96,97,98,99,100]. The smaller pie chart is representative of a single hospital study. The world map was obtained from Freepik.com. See also Appendix A for the toxin gene profile of listed ribotypes.

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
