# Peer review of "Make It Less difficile: Understanding Genetic Evolution and Global Spread of Clostridioides difficile"

_genes, 2022, doi:10.3390/genes13122200_

Round 1

Reviewer 1 Report

The manuscript is an interesting review on Clostridioides difficile, its genetic evolution and future perspectives.

The English is good along the manuscript and there are no special faults in terms of content.

Author Response

We would like to thank the Reviewer for reading our manuscript carefully and for appreciating our work.

Reviewer 2 Report

Thank you for the opportunity to review this manuscript. Overall, it provides a nice summary of the global distribution of C. difficile and points out where the data is still lacking. However, I believe there are a few sections that may need to be clarified.

Major issues:

1.      I disagree with how the authors describe AMR in C. difficile (Lines 157 – 201). Resistance to antimicrobials for CDI treatment is rare. In Minor point #2 below, only 0.07% of C. difficile has metronidazole MIC above ECOFF. Most countries in the Pan-European study (PMID: 25701178) reported metronidazole, vancomycin and fidaxomicin resistance below 10% (most of them reported 0%). A species-wide genomic study (PMID: 34793295) could identify pCD-METRO plasmid in only 15 strains (out of > 10,000 strains), most of which were from the original pCD-METRO study. The real problem with AMR in C. difficile is the resistance that helps with the spread of C. difficile, like fluoroquinolone resistance in RT027, tetracycline resistance in RT078 and clindamycin resistance in RT017.

2.      The lack of hypervirulent strains keeps popping up in this review (for example, in Line 380), but the authors did not discuss any possible explanation (like the lack of AMR compared to the developed countries). This could be a good selling point of this paper.

Minor issues:

3.      Lines 33 – 34: This is a nice summary of the pathogenesis of CDI. It would be better if the authors can also include bits about the roles of microbiota (e.g. the conversion of bile acids) and how antimicrobial treatment can induce CDI.

4.      Lines 79 – 82: What does this mean? Of course, there must be some isolate with MIC more than the ECOFF limit, but it does not indicate that there is a high prevalence of resistance. For instance, of 7842 C. difficile isolates, only 56 have metronidazole MIC over the ECOFF. That is less than 1%.

5.      Line 119: Why do the authors need to specifically include this strain? Based on reference [39] the conserved 115 bp region has been known for some time.

6.      Line 194: Tigecycline does not belong to the tetracycline class, but rather glycylcyclines, and I do not think the tet genes confer resistance to this drug.

7.      When describing ribotypes, it may be better to also include the toxin gene profiles (e.g. RT027[A+B+CDT+]) to help the readers.

8.      For Europe and the US, it may be nice to also discuss the disappearance of RT027. The US CDC provides a nice description of how fluoroquinolone control led to a decrease in RT027 cases and overall mortality.

9.      Line 311: This is my personal perception (although factually wrong). When I read Latin America, I think of countries in Central America. As the authors only described studies from Brazil. It may be better to refer to the region as South America.

10.   Lines 324 – 332: Instead of referring to the countries as low- and middle-income countries, it may be better to classify the countries based on the geographical regions. There is a lot of information in East Asia (China and Japan) and Southeast Asia, but very little in South Asia.

11.   Line 349: The authors mentioned the vastness of Australia to explain the diversity of C. difficile in the country. However, the authors compared a study in Perth (RT002 and RT009) to a study in Western Australia (RT014/020). Perth is also in Western Australia. I don’t think the geographical area is not an issue here, but rather the difference in the tested population.

12.   Line 370: (Very minor) Maybe put RT014 before RT020 for consistency.

Author Response

Thank you for the opportunity to review this manuscript. Overall, it provides a nice summary of the global distribution of C. difficile and points out where the data is still lacking. However, I believe there are a few sections that may need to be clarified.

We are very pleased that the Reviewer has carefully evaluated and appreciated our work, and we hope that the revised version of our manuscript satisfies the critical issues encountered.

Major issues:

  1. I disagree with how the authors describe AMR in C. difficile(Lines 157 – 201). Resistance to antimicrobials for CDI treatment is rare. In Minor point #2 below, only 0.07% of C. difficile has metronidazole MIC above ECOFF. Most countries in the Pan-European study (PMID: 25701178) reported metronidazole, vancomycin and fidaxomicin resistance below 10% (most of them reported 0%). A species-wide genomic study (PMID: 34793295) could identify pCD-METRO plasmid in only 15 strains (out of > 10,000 strains), most of which were from the original pCD-METRO study. The real problem with AMR in C. difficile is the resistance that helps with the spread of C. difficile, like fluoroquinolone resistance in RT027, tetracycline resistance in RT078 and clindamycin resistance in RT017.

The Reviewer is right and we apologize for the inaccuracy on this point. In the revised version of our manuscript, the paragraph on antibiotic resistance in C. difficile has been implemented by providing more detail on the real issue of AMR in C. difficile leading to the spread of multidrug-resistant C. difficile strains, and discussing how the guidelines for antimicrobial treatment have changed over time (lines 180-193). Although resistance to antimicrobials for CDI treatment is indeed rare, we considered it appropriate to keep the part on antimicrobial resistance mechanisms for commonly used compounds, for narrative consistency. However, if the Reviewer deems it appropriate to further amend or remove this paragraph, we will be more than willing to accept suggestions.

  1. The lack of hypervirulent strains keeps popping up in this review (for example, in Line 380), but the authors did not discuss any possible explanation (like the lack of AMR compared to the developed countries). This could be a good selling point of this paper.

We would like to thank the Reviewer for the valuable suggestion.

In the revised version of our manuscript, we have implemented the Conclusions section by putting forward some hypotheses in support of the absence of hypervirulent strains, especially in some countries (lines 516-525). For example, it has been suggested that low availability of antimicrobials or co-infection with other pathogens (such as the causative agents of tuberculosis and malaria) affect (reducing) the virulence of C. difficile strains in Asia and Africa. We have modified the end of the Introduction section accordingly, anticipating that we will discuss this aspect (lines 123-124).  

Minor issues:

  1. Lines 33 – 34: This is a nice summary of the pathogenesis of CDI. It would be better if the authors can also include bits about the roles of microbiota (e.g. the conversion of bile acids) and how antimicrobial treatment can induce CDI.

We are in complete agreement with the Reviewer and thank her/him once again for the valuable suggestions.

In the revised version of our manuscript, we have implemented the Introduction section by delving into the roles of the gut microbiota in CDI, including its bile acid-modifying activity, and discussing how antibiotic treatment-induced dysbiosis can contribute to CDI (line 68-84).

  1. Lines 79 – 82: What does this mean? Of course, there must be some isolate with MIC more than the ECOFF limit, but it does not indicate that there is a high prevalence of resistance. For instance, of 7842 C. difficileisolates, only 56 have metronidazole MIC over the ECOFF. That is less than 1%.

The Reviewer is right, and we apologize for the inaccuracy.

As suggested, we have modified the Introduction section by clarifying that only in some cases the MIC is above the ECOFF limit (line 96) and specifying that resistance to antimicrobials recommended by the guidelines is fortunately rare (lines 99-101).

  1. Line 119: Why do the authors need to specifically include this strain? Based on reference [39] the conserved 115 bp region has been known for some time.

The Reviewer is right, we included this information in order to make the paragraph more comprehensive and argue for the lack of toxin genes in non-toxigenic C. difficile strains (please, see lines 139-140). However, if the Reviewer finds this part irrelevant, we will be more than willing to delete it.

  1. Line 194: Tigecycline does not belong to the tetracycline class, but rather glycylcyclines, and I do not think the tetgenes confer resistance to this drug.

We apologize again for the inaccuracy. As correctly stated by the Reviewer, tigecycline is an antibiotic belonging to the group of glycylcyclines, structurally similar to tetracyclines [1]. We have modified the main text by adding this specification (lines 228-229). As for the tet genes, in the literature it appears that these genes can confer resistance to tigecycline in various microorganisms of clinical relevance [2,3]; in the revised version of our manuscript, we have briefly discussed this point (lines 231-235).

  1. When describing ribotypes, it may be better to also include the toxin gene profiles (e.g. RT027[A+B+CDT+]) to help the readers.

We thank the Reviewer for this valuable suggestion.

In the revised version of our manuscript, we have added a new Table S1, which shows the toxin gene profile (i.e., presence/absence of tcd and ctd genes) of the RTs mentioned in the main text (lines 287-288). See also lines 1099.

  1. For Europe and the US, it may be nice to also discuss the disappearance of RT027. The US CDC provides a nice description of how fluoroquinolone control led to a decrease in RT027 cases and overall mortality.

Thanks again for the suggestion!

In the revised version of our manuscript, we have discussed the correlation between fluoroquinolone administration and CDI outbreaks with RT027 (lines 310-315).

  1. Line 311: This is my personal perception (although factually wrong). When I read Latin America, I think of countries in Central America. As the authors only described studies from Brazil. It may be better to refer to the region as South America.

As suggested, we have replaced the term “Latin America” with “South America” (see lines 351 and 356).

  1. Lines 324 – 332: Instead of referring to the countries as low- and middle-income countries, it may be better to classify the countries based on the geographical regions. There is a lot of information in East Asia (China and Japan) and Southeast Asia, but very little in South Asia.

We thank the Reviewer for pointing out this aspect and apologize for the use of inappropriate terms. We have changed the text accordingly (lines 366-367 and 511-512).

  1. Line 349: The authors mentioned the vastness of Australia to explain the diversity of C. difficile in the country. However, the authors compared a study in Perth (RT002 and RT009) to a study in Western Australia (RT014/020). Perth is also in Western Australia. I don’t think the geographical area is not an issue here, but rather the difference in the tested population.

We thank the Reviewer for raising this very interesting point.

We have modified the entire paragraph on Oceania, specifying that most of the data come from Australia, particularly from Western Australia, and that the discrepancies are possibly related to the different ages of the sampled patients (lines 389-390,392-393,395-397,399-401).

  1. Line 370: (Very minor) Maybe put RT014 before RT020 for consistency.

The change has been made (please, see line 413).

References

  1. Fang, L.X.; Chen, C.; Cui, C.Y.; Li, X.P.; Zhang, Y.; Liao, X.P.; Sun, J.; Liu, Y.H. Emerging High-Level Tigecycline Resistance: Novel Tetracycline Destructases Spread via the Mobile Tet(X). Bioessays 2020, 42, doi:10.1002/BIES.202000014.
  2. Sun, J.; Chen, C.; Cui, C.Y.; Zhang, Y.; Liu, X.; Cui, Z.H.; Ma, X.Y.; Feng, Y.; Fang, L.X.; Lian, X.L.; et al. Plasmid-Encoded Tet(X) Genes That Confer High-Level Tigecycline Resistance in Escherichia Coli. Nature Microbiology 2019 4:9 2019, 4, 1457–1464, doi:10.1038/s41564-019-0496-4.
  3. He, T.; Wang, R.; Liu, D.; Walsh, T.R.; Zhang, R.; Lv, Y.; Ke, Y.; Ji, Q.; Wei, R.; Liu, Z.; et al. Emergence of Plasmid-Mediated High-Level Tigecycline Resistance Genes in Animals and Humans. Nature Microbiology 2019 4:9 2019, 4, 1450–1456, doi:10.1038/s41564-019-0445-2.

Reviewer 3 Report

The review by Mengoli et al. aims to provide a state-of-the-art overview of current typing techniques and the global distribution of the most common C. difficile ribotypes. In addition, genetic variations in C. difficile pathogenic genes and their relationship to the clinical issues of CDI are discussed in detail. This review is well written and well structured, and I have no criticism to make. Just a small suggestion to make Figure 1 easier to read: add a table for each country.

Author Response

We really thank the Reviewer for appreciating our work.

As also suggested by Reviewer #2, the revised version of our manuscript now includes an additional table (Table S1), which summarizes the RTs cited in the main text and in the figure, and the corresponding toxigenic profiles. This table is mentioned in the main text (see lines 287-288) and in the caption of Figure 1 (see lines 1099). Although this does not entirely correspond to what the Reviewer requested, we believe it provides important additional information, giving the reader an immediate idea of the geographical distribution of toxigenic and non-toxigenic RTs.

Round 2

Reviewer 2 Report

Thank you again for the opportunity to review the revised version of this manuscript.

The authors have responded to all the comments and provided adequate explanations for each point. I believe the manuscript is suitable for publication.